# OpenReview forum: "Federated Learning for Decentralized Scientific Collaboration: Privacy-Preserving Multi-Agent AI for Cross-Domain Research"
_ICLR.cc/2025/Workshop/AgenticAI — ICLR 2025 Workshop AgenticAI Reject_

### Official Review · Reviewer_7PXz · 2025-02-28
**Lack of theoretical depth and real-world validation**

**Rating:** 5
**Confidence:** 3

**Review:**

Summary of Contributions
This paper proposes a federated learning (FL) framework tailored for decentralized scientific collaboration, featuring: 1. Multi-Agent Orchestration: Domain-specific coordinators (genomics, climate, etc.) that mediate cross-institutional model updates while resolving conflicts. 2. Privacy-Preserving Techniques: Integration of secure aggregation and differential privacy, reducing membership inference risks to "Low" (Table 1). 3. Empirical Validation: Demonstrates 35% faster convergence versus single-domain training across genomics, medical imaging, and climate forecasting tasks.

Strengths
1. Domain-Specific Customization: The multi-agent orchestrator (Section 4.4) effectively bridges heterogeneous scientific domains, outperforming vanilla FedAvg by 2–3% in accuracy (Table 1).
2. Practical Relevance: Addresses critical barriers to scientific collaboration (privacy regulations, data silos) with clear industry applications (Section 2).
3. Comprehensive Evaluation: Tests on three distinct domains (genomics, medical imaging, climate) with rigorous privacy leakage metrics.

Weaknesses and Recommendations
1. Theoretical Gaps: While methodology is well-described, the paper lacks a formal analysis of convergence guarantees or stability under domain shift. Deriving bounds for multi-agent FL equilibria would strengthen theoretical rigor.
2. Limited Real-World Testing: Experiments rely on simulated nodes (Section 5.1). A pilot deployment (e.g., hospital networks) is needed to validate networking and governance assumptions.
3. Scalability Oversights: No discussion of hierarchical FL or adaptive update protocols for scaling beyond 100+ institutions, a noted concern in prior work.

Suggested Improvements:
1. Add a theoretical section analyzing convergence under domain heterogeneity.
2. Include latency/power metrics for real-world deployment feasibility.
3. Benchmark against Bayesian FL methods to address scalability.

---

### Official Review · Reviewer_HuLu · 2025-02-28
**Not good enough.**

**Rating:** 3
**Confidence:** 4

**Review:**

This paper works on a federated learning framework for AI-driven scientific collaboration across geographically dispersed institutions. Instead of relying on centralized models or pooled datasets, the proposed approach enables distributed
scientific agents to train AI models while preserving local data privacy. Overall, the problem is innovative but the paper is not in good quality and the experiments and the analysis are not sufficient to prove the effectiveness of the proposed method.


quality: The paper is not fully developed with no clear clarification about the detailed explanation of the methodology. The experiments cover different scientific domains which are good but the results are not sufficient with no specific analysis to show the effectiveness of the proposed framework.

clarity: The paper highlights the challenges of applying federated learning in multi-agent systems for knowledge sharing and learning while maintaining privacy. However, it lacks a clear description of the proposed framework, as well as details on the experimental setup and evaluation process. Additionally, the experiments are incomplete and insufficient to demonstrate the effectiveness of the proposed framework.

originality: The problem and methodologies proposed in this paper are novel and important for multi-agent knowledge sharing and learning while preserving privacy.

Pros:
1. The problem of using federated learning to enhance knowledge learning across agents is important and the methodologies shown in this paper are novel.
2. The tasks and datasets in this paper show a diversity of scientific domains.

Cons:
1. The methodology behind this paper is not clearly presented.
2. The experimental results are not sufficient enough to show the effectiveness of the proposed framework.
3. The paper is not well-written and can be further improved.

---

### Official Review · Reviewer_7MrB · 2025-03-01
**Insufficient experimental detail, lack of theoretical rigor concerning convergence, minimal elaboration on domain-specific adaptations, and writing quality that could be enhanced for clarity and formality.**

**Rating:** 2
**Confidence:** 4

**Review:**

Summary:
This paper proposes a federated learning framework tailored to multi-institution, decentralized scientific collaborations. Instead of pooling data at a central server, each institution trains models locally and shares only updates or model parameters. The authors introduce an agentic AI orchestrator that coordinates multi-agent interactions and domain-specific adaptations, aiming to accelerate learning and reduce communication overhead.

Strengths:
1. The presentation of key points is straightforward.
2. The authors analyze the potential limitations and future directions for their work.

Weaknesses:
1. This quality of writing is not good; using more formal words would make it better and concise.
2. The paper’s structure lacks clarity and coherence, omitting key experimental details. For example, the experimental setting of table 1 is unclear. What are the results for other datasets (as the authors claimed they conducted experiments on datasets of genomics, medical imaging, and climate sensors)?
3. While the approach is well-motivated and supported by empirical evidence, the paper lacks a thorough theoretical grounding regarding convergence guarantees or equilibrium behavior in multi-agent federated setups.
4. Although the multi-agent orchestration is described, there is little detail on how domain-specific tasks or domain shifts are handled in practice. For example, medical imaging and genomics can differ considerably in data dimensionality, labeling approaches, or performance metrics. More elaboration on domain adaptations, conflict resolution strategies, or how to weigh domain priorities would be beneficial.

---

### Decision · Program_Chairs · 2025-03-05

Reject